# Trends in U.S. self-reported health and self-care behaviors during the COVID-19 pandemic

Madison Hooper[1], Morgan Reinhart[2], Stacie B. Dusetzina[3], Colin Walsh[4,5,6], Kevin N. Griffith[3,7]*

**1** Department of Psychology, Vanderbilt University, Nashville, TN, United States of America, **2** Department of Psychology & Human Development, Vanderbilt University, Nashville, TN, United States of America, **3** Department of Health Policy, Vanderbilt University Medical Center, Nashville, TN, United States of America, **4** Department of Biomedical Informatics, Vanderbilt University Medical Center, Nashville, TN, United States of America, **5** Department of Medicine, Vanderbilt University Medical Center, Nashville, TN, United States of America, **6** Department of Psychiatry and Behavioral Sciences, Vanderbilt University Medical Center, Nashville, TN, United States of America, **7** Partnered Evidence-Based Policy Resource Center, VA Boston Healthcare System, Boston, MA, United States of America

* kevin.griffith@vumc.org

**Data Availability Statement:** Our analytic dataset and R scripts are available within an open access Mendeley Data Repository, accessible at https://data.mendeley.com/datasets/nfvpcd9wpf.

## Abstract

### Importance

The COVID-19 pandemic represents a unique stressor in Americans' daily lives and access to health services. However, it remains unclear how the pandemic impacted perceived health status and engagement in health-related behaviors.

### Objective

To assess changes in self-reported health outcomes during the COVID-19 pandemic, and to explore trends in health-related behaviors that may underlie the observed health changes.

### Design

Interrupted time series stratified by age, gender, race/ethnicity, educational attainment, household income, and employment status.

### Setting

United States.

### Participants

All adult respondents to the 2016–2020 Behavioral Risk Factor Surveillance System (N = 2,146,384).

### Exposure

Survey completion following the U.S. public health emergency declaration (March-December 2020). January 2019 to February 2020 served as our reference period.

**Funding:** KNG's effort was supported in part by a grant from the U.S. Agency for Healthcare Research & Quality (https://www.ahrq.gov/, K12 HS026395). There was no additional external funding received for this study. The funders had no role in study design, data collection and analysis, decision to publish, or preparation of the manuscript.

**Competing interests:** The authors have declared that no competing interests exist.

## Main outcomes and measures

Self-reported health outcomes included the number of days per month that respondents spent in poor mental health, physical health, or when poor health prevented their usual activities of daily living. Self-reported health behaviors included the number of hours slept per day, number of days in the past month where alcohol was consumed, participation in any exercise, and current smoking status.

## Results

The national rate of days spent in poor physical health decreased overall (-1.00 days, 95% CI: -1.10 to -0.90) and for all analyzed subgroups. The rate of poor mental health days or days when poor health prevented usual activities did not change overall but exhibited substantial heterogeneity by subgroup. We also observed overall increases in mean sleep hours per day (+0.09, 95% CI 0.05 to 0.13), the percentage of adults who report any exercise activity (+3.28%, 95% CI 2.48 to 4.09), increased alcohol consumption days (0.27, 95% CI 0.18 to 0.37), and decreased smoking prevalence (-1.11%, 95% CI -1.39 to -0.83).

## Conclusions and relevance

The COVID-19 pandemic had deleterious but heterogeneous effects on mental health, days when poor health prevented usual activities, and alcohol consumption. In contrast, the pandemic's onset was associated with improvements in physical health, mean hours of sleep per day, exercise participation, and smoking status. These findings highlight the need for targeted outreach and interventions to improve mental health in individuals who may be disproportionately affected by the pandemic.

## Introduction

The novel coronavirus (COVID-19) began to spread across the world in December 2019 and on March 13, 2020, the United States declared a national public health emergency [1,2]. Efforts to mitigate the spread of the virus included social distancing, stay-at-home orders, closure of nonessential business, travel restrictions, and quarantines [3]. Additionally, the Centers for Disease Control (CDC) and Centers for Medicare and Medicaid Services advised hospitals across the United States to prioritize urgent visits and delay elective or noncritical medical services to diminish the spread of COVID-19 in health care settings [4,5]. According to a Kaiser Family Foundation poll conducted in May 2020, 48% of U.S. adults reported that they or someone in their household skipped or postponed medical care because of the virus, and 11% of them reported that their health deteriorated as a result [6,7].

Anecdotal evidence for the pandemic's deleterious impacts on mental health have been covered widely in national media, and survey research suggests changes in mental health both overall and for distinct population subgroups [8–11]. For instance, 40% of U.S. adults reported that stress related to COVID-19 has negatively impacted their mental health [6,7]. However, these findings relied on the Census Bureau's Household Pulse Survey which had very low response rates nearing 7%. Young adults, college students, low-income households, and Hispanics were also more likely to report psychological distress during the pandemic compared to other population subgroups [8,11].

Prior work on the pandemic's health impacts generally focused on the acute or long-run health effects of COVID-19 infection. To our knowledge, there has been no prior work

documenting population-level health effects during the pandemic. Certain features of the pandemic may mitigate against negative health impacts due to deferred/avoided care. Prior work suggests that individuals might adopt healthier lifestyles during periods of unemployment or reduced economic activity [12]. Additionally, the pandemic triggered a recession that may lead to concomitant reductions in job-related stress and hazardous working conditions for some adults [12–14]. Thus, the net effect of the pandemic on *population-level* physical and mental health is unclear. We addressed these gaps and assessed overall changes in population-level mental and physical health during the COVID-19 pandemic, described changes in the mental and physical health of key population subgroups, and explored related trends in engagement in health-related behaviors (sleep and exercise) that may underlie the observed changes in health outcomes.

## Methods

### Study sample

Data for this study were obtained from the 2016–2020 Behavioral Risk Factor Surveillance System (BRFSS). The BRFSS is an annual telephone survey of approximately 450,000 households that is conducted by the states in partnership with the CDC. Respondents provide information on their health behaviors, physical and mental health conditions, use of health services, and demographic characteristics. The BRFSS includes weights to account for differential rates of survey non-response across varying population subgroups, thereby ensuring the weighted sample is representative of the U.S. adult noninstitutionalized population.

### Study variables

We examined seven self-reported outcomes for individual's health status and health behaviors. These included the number of days during the past month (range 0–30) that the respondent spent in poor mental health, poor physical health, or when poor health prevented their usual daily activities. We also examined the number of hours slept per day (range 0–24), number of days where alcohol was consumed during the past month (range 0–30), participation in any exercise within the past month (coded as 0 if no, 1 if yes), and smoking status (coded as 0 if non-smoker, 1 otherwise). The exact text and response format of each survey question are listed in **Table 1**.

Self-reported covariates included binary indicators for respondent sex, primary race (categorized in the BRFSS as White non-Hispanic, Black non-Hispanic, Other race non-Hispanic, Multiracial non-Hispanic, Hispanic), household income category (less than $15,000, $15,000 to $25,000, $25,000 to $35,000, $35,000 to $50,000, $50,000 to $75,000, and over $75,000), age category (18 to 24, 25 to 34, 35 to 44, 45 to 54, 55 to 64, and 65-plus), veteran status, educational attainment (less than college graduate, current college student, college graduate), whether the respondent is currently married, pregnancy status, presence of children in the household, whether the respondent owns their home, and employment status (categorized as employed, unemployed, or not in the labor force). We also included dummy variables for survey language (English or non-English), whether the survey was conducted via cellphone or land line, and state of residence. We excluded potential covariates that may change due to the pandemic's onset (e.g., insurance coverage, household income), as these may bias our effect estimates.

### Analytic approach

Our analysis proceeded in three steps. First, we estimated unadjusted means for each calendar month to illuminate trends in outcomes throughout the study period. Second, we estimated

**Table 1. List of study outcomes.**

| Outcome Name | Survey Question | Response Format |
|---|---|---|
| Mental Health (N = 2,108,270) | "For how many days during the past 30 days was your mental health not good?" | 0–30 |
| Physical Health (N = 2,099,610) | "For how many days during the past 30 days was your physical health not good?" | 0–30 |
| Poor Health (N = 2,119,907) | "During the past 30 days, for about how many days did poor physical or mental health keep you from doing your usual activities, such as self-care, work, or recreation?" | 0–30 |
| Sleep Hours[1] (N = 1,326,579) | "On average, how many hours of sleep do you get in a 24-hour period?" | 0–24 |
| Exercise Participation (N = 2,088,459) | "During the past month, other than your regular job, did you participate in any physical activities or exercises such as running, calisthenics, golf, gardening, or walking for exercise?" | Yes/No |
| Alcohol Consumption (N = 2,026,525) | "During the past 30 days, how many days per week or per month did you have at least one drink of any alcoholic beverage such as beer, wine, a malt beverage or liquor?" | Week: 0–7[2] Month: 0–30 |
| Smoking Status (N = 2,052,550) | "Do you now smoke cigarettes every day, some days, or not at all?" | Every day, Some days, Not at all[3] |

[1]This question is not asked in most states during odd years and was excluded from the 2019 BRFSS

[2]Converted to monthly for consistency

[3]Binary coded where 1 = every day or some days, 0 = not at all.

covariate-adjusted interrupted time series regression models to identify overall changes in self-reported health before and after the pandemic's onset. We included dummy variables for whether each survey response was collected in 2016, 2017, 2018. Our key variable of interest was a binary indicator taking on a value of one if the response was collected after the US public health emergency declaration (March 2020 to December 2020), 0 otherwise. Thus, January 2019 to February 2020 served as our reference period. Due to the large sample size and our interest in population average effects, we opted to conduct linear probability models using ordinary least squares to aid in interpretability of results [15].

Third, we re-estimated our regression models stratified into one of 26 different population subgroups based on sex, income, race/ethnicity, age, educational attainment, employment status, or student status. We employed heteroskedasticity-robust standard errors in all regression models. Errors were also clustered by state to account for pandemic-related policies that may affect their residents. All data preparation and analyses were conducted in Microsoft R Open version 4.0.2. Our analytic dataset and R scripts are available within an open access Mendeley Data Repository, accessible at https://data.mendeley.com/datasets/nfvpcd9wpf. This project was considered exempt by the Vanderbilt University Medical Center Institutional Review Board. Informed consent was waived because the study involved only secondary analyses of existing data.

## Results

Our final sample included 2,146,384 unweighted survey respondents. After weighting, our sample demographics were reflective of the U.S. general population (**Table 2**). Covariate-level missingness ranged from 0–20%; we used multiple imputation using additive regression, bootstrapping, and predictive mean matching to reduce the potential for non-response bias [16,17]. Respondents with missing data for individual outcomes were excluded for those specific analyses. Baseline rates for all outcomes during our pre-pandemic reference period (January 2019 to February 2020) are contained in **Table 3**.

**Table 2. Characteristics of the Study Sample (N = 2,146,384).**

| Variable | | N | Weighted %[1] |
|---|---|---|---|
| Sex | | | |
| | Female | 1,185,025 | 51.3 |
| | Male | 961,359 | 48.7 |
| Income Group | | | |
| | Less than $15,000 | 206,912 | 10.8 |
| | $15,000 to $25,000 | 353,727 | 16.9 |
| | $25,000 to $35,000 | 227,076 | 10.3 |
| | $35,000 to $50,000 | 301,525 | 13.2 |
| | $50,000 to $75,000 | 340,247 | 14.8 |
| | $75,000 + | 716,897 | 34.1 |
| Race | | | |
| | White | 1,665,833 | 63.5 |
| | Black | 172,350 | 11.8 |
| | Hispanic | 159,701 | 16.2 |
| | Other race | 105,897 | 7.2 |
| | Multiracial | 42,603 | 1.4 |
| Age | | | |
| | 18 to 24 | 127,310 | 12.4 |
| | 25 to 34 | 227,259 | 17.5 |
| | 35 to 44 | 254,871 | 16.2 |
| | 45 to 54 | 318,698 | 15.8 |
| | 55 to 64 | 448,482 | 16.8 |
| | 65+ | 769,764 | 21.3 |
| Education Group | | | |
| | College grad | 809,004 | 72.0 |
| | Not a college grad | 1,337,380 | 28.0 |
| Employment Group | | | |
| | Unemployed | 92,807 | 5.7 |
| | Employed | 1,080,504 | 57.4 |
| | Not in labor force | 973,073 | 36.9 |
| Student Status | | | |
| | Student | 56,756 | 5.5 |
| | Not a student | 2,089,628 | 94.5 |

**Source:** Authors' analysis of data from the 2016–2020 Behavioral Risk Factor Surveillance System (BRFSS). **Notes**
[1] Incorporates BRFSS post-stratification weights.

## Changes in self-reported health

Trends in self-reported health outcomes and engagement in health behaviors were generally stable prior to the onset of the COVID-19 pandemic (Fig 1, Appendix A2-A5 in S1 File). In adjusted regression models, the national rate of days spent in poor mental health did not change significantly during the pandemic (-0.03 days, 95% CI: -0.16 to 0.09) (Table 4). How-ever, certain population subgroups experienced significant increases in poor mental health days; the largest increases were observed for respondents who were not college graduates (+0.39 days, 95% CI: 0.31 to 0.47), lived in households earning more than $75,000 per year (+0.29 days, 95% CI: 0.20 to 0.37), or were currently employed (+0.11 days, 95% CI: 0.02 to 0.20). In contrast, days spent in poor mental health decreased among those who were

**Table 3. Baseline outcomes prior to the onset of the COVID-19 pandemic.**

| Strata | | Poor mental health days (#) | Poor physical health days (#) | Days when health prevented activities (#) | Sleep hours per day (#) | Any exercise during the past month (%) | Days when alcohol was consumed (#) | Smoke cigarettes (%) |
|---|---|---|---|---|---|---|---|---|
| Overall National Rate | | 4.30 | 4.08 | 5.07 | 6.97 | 73.7 | 4.65 | 15.4 |
| Sex | | | | | | | | |
| | Female | 4.90 | 4.41 | 5.12 | 7.01 | 71.9 | 3.53 | 13.5 |
| | Male | 3.67 | 3.72 | 5.01 | 6.93 | 75.5 | 5.84 | 17.3 |
| Income Group | | | | | | | | |
| | Less than $15,000 | 6.92 | 7.63 | 9.14 | 6.99 | 60.0 | 2.51 | 24.6 |
| | $15,000 to $25,000 | 5.58 | 5.77 | 6.91 | 6.92 | 63.3 | 3.07 | 21.7 |
| | $25,000 to $35,000 | 4.62 | 4.51 | 5.18 | 7.04 | 67.8 | 3.73 | 18.4 |
| | $35,000 to $50,000 | 4.37 | 3.87 | 4.69 | 6.96 | 72.4 | 4.27 | 16.0 |
| | $50,000 to $75,000 | 3.82 | 3.29 | 3.97 | 6.94 | 76.7 | 5.01 | 13.8 |
| | $75,000 + | 2.99 | 2.51 | 3.05 | 6.98 | 83.5 | 6.28 | 9.2 |
| Race | | | | | | | | |
| | White | 4.37 | 4.21 | 5.09 | 7.00 | 75.6 | 5.39 | 15.8 |
| | Black | 4.54 | 4.10 | 5.34 | 6.78 | 68.8 | 3.55 | 17.8 |
| | Hispanic | 3.98 | 3.89 | 4.96 | 6.96 | 67.7 | 3.20 | 12.8 |
| | Other race | 3.63 | 3.16 | 4.48 | 7.02 | 76.5 | 3.14 | 11.9 |
| | Multiracial | 6.41 | 5.01 | 5.84 | 6.72 | 77.5 | 4.30 | 22.6 |
| Age | | | | | | | | |
| | 18 to 24 | 6.34 | 2.41 | 3.52 | 6.96 | 80.4 | 3.59 | 10.3 |
| | 25 to 34 | 5.25 | 2.71 | 3.76 | 6.84 | 77.4 | 4.89 | 18.9 |
| | 35 to 44 | 4.47 | 3.25 | 4.31 | 6.78 | 75.7 | 4.83 | 19.7 |
| | 45 to 54 | 4.21 | 4.42 | 5.78 | 6.85 | 72.6 | 4.78 | 17.4 |
| | 55 to 64 | 3.91 | 5.50 | 6.87 | 6.91 | 71.0 | 4.90 | 17.7 |
| | 65+ | 2.63 | 5.38 | 6.14 | 7.30 | 68.4 | 4.66 | 9.2 |
| Education Group | | | | | | | | |
| | College grad | 3.07 | 2.65 | 3.29 | 7.04 | 85.2 | 6.13 | 6.1 |
| | Not a college grad | 4.78 | 4.64 | 5.73 | 6.94 | 69.1 | 4.06 | 19.1 |
| Employment Group | | | | | | | | |
| | Unemployed | 7.36 | 5.68 | 8.05 | 6.92 | 70.0 | 4.04 | 27.0 |
| | Employed | 3.74 | 2.51 | 2.90 | 6.84 | 77.1 | 5.32 | 15.4 |
| | Not in labor force | 4.75 | 6.32 | 7.63 | 7.16 | 68.9 | 3.72 | 13.8 |
| Student Status | | | | | | | | |
| | Student | 5.99 | 2.21 | 3.36 | 7.05 | 85.2 | 3.01 | 5.6 |
| | Not a student | 4.20 | 4.18 | 5.20 | 6.96 | 73.0 | 4.75 | 15.9 |

**Source:** Authors' analysis of data from the 2016–2020 Behavioral Risk Factor Surveillance System (BRFSS). **Notes:** The exhibit displays mean outcomes during our pre-pandemic baseline period of January 2019 through February 2020, after accounting for BRFSS post-stratification weights.

unemployed (-1.00 days, 95% CI: -1.39 to -0.60), had household incomes between $15,000 to $25,000 (-0.39 days, 95% CI: -0.68 to -0.11), college graduates (-0.21 days, 95% CI: -0.37 to -0.05), or those aged 18–24 (-0.19 days, 95% CI: -0.31 to -0.06).

The number of days spent in poor physical health significantly decreased after the start of the pandemic (-1.00 days, 95% CI: -1.10 to -0.90). After stratifying by demographics, we found that physical health improved significantly (i.e., number of days spent in poor physical health declined) for every single demographic group after the start of the pandemic. The groups that

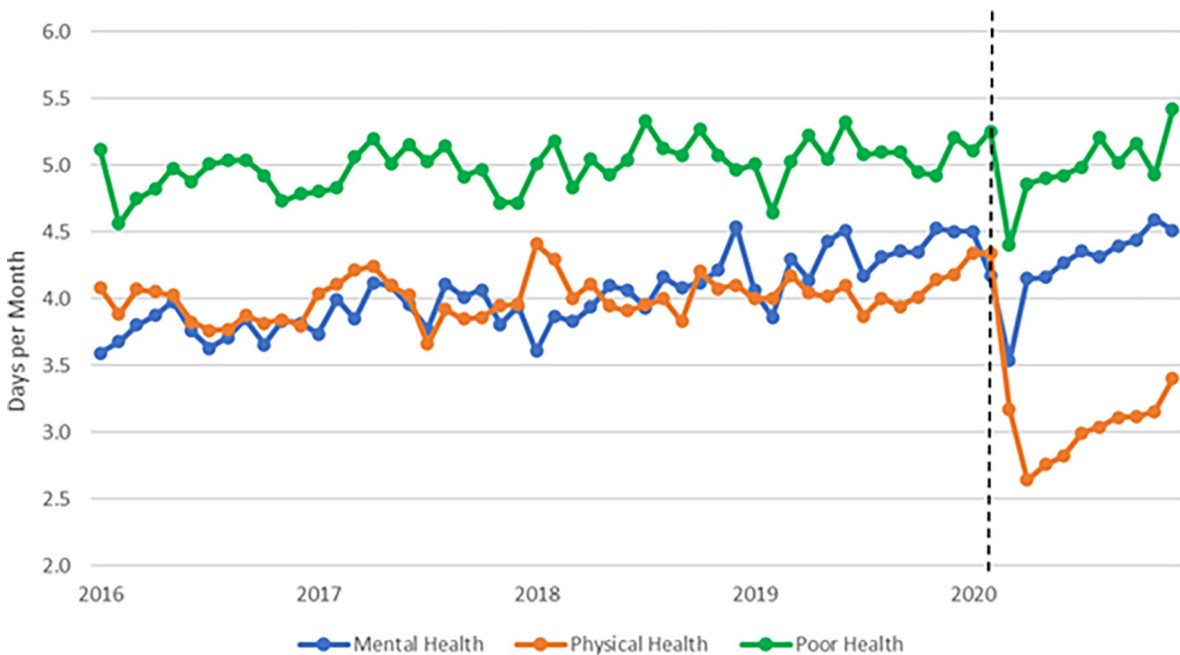

**Fig 1. National trends in poor physical and mental health days, 2016–2020. Source:** Authors' analysis of data from the 2016–2020 BRFSS. **Notes:** The figure displays monthly unadjusted trends in outcomes, accounting for BRFSS post-stratification weights. The vertical dashed line indicates February 2023.

reported the largest improvements were those who were unemployed (-2.05 days, 95% CI: -2.32 to -1.77), not in the labor force (-1.30 days, 95% CI: -1.47 to -1.14), and those in households earning an income less than $15,000 (-1.57 days, 95% CI: -1.87 to -1.28) or between $15,000 and $25,000 (-1.26 days, 95% CI: -1.56 to -0.95).

Nationally, the number of days when poor health prevented usual activities did not change significantly after the start of the pandemic (-0.03 days, 95% CI: -0.12 to 0.06). However, several groups reported significant increases in poor health days including students (+0.70 days, 95% CI: 0.30 to 1.10), those aged 18–24 (+0.36 days, 95% CI: 0.12 to 0.61) or 25–34 (+0.28 days, 95% CI: 0.09 to 0.46), Hispanic respondents (+0.38 days, 95% CI: 0.04 to 0.73), and households earning between $25,000 and $35,000 per year (+0.58 days, 95% CI: 0.11 to 1.06). In contrast, we observed significant decreases in the number of poor health days were those who were unemployed (-1.39 days, 95% CI: -1.68 to -1.09) or not in the labor force (-0.35 days, 95% CI: -0.50 to -0.21), White respondents (-0.22 days, 95% CI: -0.32 to -0.12), those aged 65+ (-0.24 days, 95% CI: -0.48, -0.00), and households earning between $15,000 and $25,000 (-0.38 days, 95% CI: -0.67 to -0.09).

## Changes in self-reported health behaviors

The adjusted regression models revealed that Americans' mean hours of sleep per day significantly increased after the pandemic's onset (+0.09 hours, 95% CI: 0.05 to 0.13) (**Table 5**). Most population groups experienced significant increases in sleep hours per day; the largest increases were observed for those aged 18 to 24 years (+0.21 hours, 95% CI: 0.11 to 0.31), the unemployed (+0.18 hours, 95% CI: 0.07 to 0.29), households earning between $15,000 and $25,000 (+0.17, 95% CI: 0.07 to 0.27), or Black respondents (+0.15 hours, 95% CI: 0.05 to 0.26). In contrast, mean sleep hours did not change significantly for respondents in households earning between $25,000 and $35,000, those who identified as multiracial or other race, and those aged 45+.

**Table 4. Adjusted regression estimates for changes in poor health days during the COVID-19 pandemic.**

| Variable | | Poor mental health days (#) | | Poor physical health days (#) | | Days when poor health prevented activities (#) | |
|---|---|---|---|---|---|---|---|
| | | ß | 95% CI | ß | 95% CI | ß | 95% CI |
| Overall National Rate | | -0.03 | (-0.16, 0.09) | -1.00*** | (-1.10, -0.90) | -0.03 | (-0.12, 0.06) |
| Sex | | | | | | | |
| | Female | 0.11 | (-0.05, 0.28) | -1.08*** | (-1.17, -0.98) | 0.03 | (-0.10, 0.16) |
| | Male | -0.19** | (-0.31, -0.06) | -0.92*** | (-1.04, -0.79) | -0.11 | (-0.24, 0.03) |
| Income Group | | | | | | | |
| | Less than $15,000 | -0.30 | (-0.67, 0.06) | -1.57*** | (-1.87, -1.28) | -0.03 | (-0.45, 0.39) |
| | $15,000 to $25,000 | -0.39** | (-0.68, -0.11) | -1.26*** | (-1.56, -0.95) | -0.38* | (-0.67, -0.09) |
| | $25,000 to $35,000 | -0.12 | (-0.35, 0.10) | -0.92*** | (-1.07, -0.77) | 0.58* | (0.11, 1.06) |
| | $35,000 to $50,000 | 0.04 | (-0.33, 0.41) | -0.90*** | (-1.13, -0.67) | 0.07 | (-0.17, 0.31) |
| | $50,000 to $75,000 | -0.02 | (-0.20, 0.16) | -0.73*** | (-0.88, -0.59) | -0.01 | (-0.27, 0.26) |
| | $75,000 + | 0.29*** | (0.20, 0.37) | -0.77*** | (-0.90, -0.64) | 0.09 | (-0.08, 0.27) |
| Race | | | | | | | |
| | White | -0.01 | (-0.13, 0.11) | -1.06*** | (-1.15, -0.96) | -0.22*** | (-0.32, -0.12) |
| | Black | -0.19 | (-0.45, 0.06) | -0.91*** | (-1.15, -0.68) | 0.32 | (-0.21, 0.85) |
| | Hispanic | 0.07 | (-0.33, 0.48) | -0.89*** | (-1.28, -0.50) | 0.38* | (0.04, 0.73) |
| | Other race | -0.22 | (-0.68, 0.24) | -0.95*** | (-1.20, -0.70) | 0.13 | (-0.24, 0.51) |
| | Multiracial | -0.06 | (-0.53, 0.40) | -0.76* | (-1.46, -0.06) | 0.30 | (-0.42, 1.01) |
| Age | | | | | | | |
| | 18 to 24 | -0.33** | (-0.58, -0.09) | -0.75*** | (-0.93, -0.56) | 0.36** | (0.12, 0.61) |
| | 25 to 34 | -0.05 | (-0.28, 0.18) | -0.78*** | (-0.95, -0.62) | 0.28** | (0.09, 0.46) |
| | 35 to 44 | 0.07 | (-0.16, 0.30) | -0.89*** | (-1.07, -0.72) | 0.02 | (-0.29, 0.33) |
| | 45 to 54 | 0.19 | (-0.02, 0.41) | -1.10*** | (-1.26, -0.93) | -0.15 | (-0.43, 0.13) |
| | 55 to 64 | -0.12 | (-0.36, 0.11) | -1.32*** | (-1.62, -1.03) | -0.39 | (-0.81, 0.03) |
| | 65+ | 0.00 | (-0.14, 0.14) | -1.02*** | (-1.22, -0.82) | -0.24* | (-0.48, 0.00) |
| Education Group | | | | | | | |
| | College grad | -0.21* | (-0.37, -0.05) | -1.07*** | (-1.24, -0.91) | -0.07 | (-0.19, 0.05) |
| | Not a college grad | 0.39*** | (0.31, 0.47) | -0.82*** | (-0.92, -0.71) | 0.08 | (-0.03, 0.20) |
| Employment Group | | | | | | | |
| | Unemployed | -1.00*** | (-1.39, -0.60) | -2.05*** | (-2.32, -1.77) | -1.39*** | (-1.68, -1.09) |
| | Employed | 0.11* | (0.02, 0.20) | -0.71*** | (-0.79, -0.64) | 0.26*** | (0.14, 0.38) |
| | Not in labor force | -0.20 | (-0.4, 0.00) | -1.30*** | (-1.47, -1.14) | -0.35*** | (-0.50, -0.21) |
| Student Status | | | | | | | |
| | Student | 0.11 | (-0.45, 0.68) | -0.54*** | (-0.74, -0.33) | 0.70*** | (0.30, 1.10) |
| | Not a student | -0.04 | (-0.15, 0.07) | -1.02*** | (-1.13, -0.92) | -0.08 | (-0.18, 0.01) |

**Source:** Authors' analysis of data from the 2016–2020 Behavioral Risk Factor Surveillance System (BRFSS). **Notes:** The exhibit displays regression-adjusted estimates for the number of days during the past 30 that were spent in poor mental health, poor physical health, or when poor health prevented usual activities. All models were accounted for BRFSS post-stratification weights, with standard errors clustered by state. Significance codes

*p< .05

**p < .01

***p< .001.

Exercise participation significantly improved after the start of the pandemic, both overall (+3.28 percentage points (pp), 95% CI: 2.48 to 4.09) and most population subgroups. The largest improvements in exercise participation accrued to the unemployed (+5.73 pp, 95% CI: 4.03 to 7.42), those between the ages of 45 and 54 years (+4.56 pp, 95% CI: 3.52 to 5.60), and White

**Table 5. Adjusted regression estimates for the changes in sleep and exercise behavior during the COVID-19 pandemic.**

| | | Sleep hours per day (#) | | Any exercise during the past month (%) | | Days consuming alcohol (#) | | Current smoker (%) | |
|---|---|---|---|---|---|---|---|---|---|
| **Variable** | | ß | 95% CI | ß | 95% CI | ß | 95% CI | ß | 95% CI |
| Overall National Rate | | 0.09*** | (0.05, 0.13) | 3.28*** | (2.48, 4.09) | 0.27*** | (0.18, 0.37) | -1.11*** | (-1.39, -0.83) |
| Sex | | | | | | | | | |
| | Female | 0.09*** | (0.05, 0.12) | 3.35*** | (2.57, 4.12) | 0.40*** | (0.31, 0.48) | -1.12*** | (-1.41, -0.82) |
| | Male | 0.09*** | (0.04, 0.15) | 3.22*** | (2.25, 4.19) | 0.15* | (0.01, 0.29) | -1.08*** | (-1.51, -0.66) |
| Income Group | | | | | | | | | |
| | Less than $15,000 | 0.13** | (0.05, 0.22) | 1.25 | (-0.48, 2.98) | 0.21** | (0.07, 0.35) | -0.76 | (-1.92, 0.40) |
| | $15,000 to $25,000 | 0.17** | (0.07, 0.27) | 4.05*** | (2.73, 5.38) | 0.16 | (-0.03, 0.34) | -1.93*** | (-2.55, -1.31) |
| | $25,000 to $35,000 | 0.04 | (-0.06, 0.14) | 3.22*** | (1.79, 4.66) | 0.11 | (-0.18, 0.40) | -1.08* | (-1.98, -0.19) |
| | $35,000 to $50,000 | 0.06** | (0.02, 0.10) | 3.35*** | (2.18, 4.52) | 0.30*** | (0.15, 0.45) | -0.30 | (-1.03, 0.42) |
| | $50,000 to $75,000 | 0.09** | (0.02, 0.16) | 2.78*** | (1.41, 4.14) | 0.25** | (0.10, 0.39) | -1.22*** | (-1.82, -0.62) |
| | $75,000 + | 0.06** | (0.02, 0.11) | 3.36*** | (2.55, 4.17) | 0.33*** | (0.17, 0.49) | -0.71*** | (-1.11, -0.31) |
| Race | | | | | | | | | |
| | White | 0.07*** | (0.04, 0.11) | 4.06*** | (3.35, 4.77) | 0.38*** | (0.27, 0.49) | -0.86*** | (-1.12, -0.59) |
| | Black | 0.15** | (0.05, 0.26) | 3.46*** | (2.58, 4.34) | 0.18 | (-0.02, 0.39) | -1.58*** | (-2.40, -0.75) |
| | Hispanic | 0.14** | (0.05, 0.24) | 1.64 | (-0.44, 3.72) | 0.11 | (-0.03, 0.26) | -2.14*** | (-3.09, -1.18) |
| | Other race | -0.03 | (-0.16, 0.11) | 0.65 | (-1.09, 2.40) | -0.05 | (-0.27, 0.17) | -0.56 | (-1.67, 0.56) |
| | Multiracial | 0.13 | (-0.02, 0.28) | 1.27 | (-0.78, 3.31) | 0.49** | (0.17, 0.80) | -1.56 | (-3.21, 0.08) |
| Age | | | | | | | | | |
| | 18 to 24 | 0.21*** | (0.11, 0.31) | 2.22** | (0.71, 3.72) | -0.02 | (-0.21, 0.17) | -1.98*** | (-2.81, -1.14) |
| | 25 to 34 | 0.10** | (0.02, 0.17) | 4.04*** | (2.66, 5.43) | 0.25* | (0.04, 0.46) | -2.34*** | (-2.92, -1.77) |
| | 35 to 44 | 0.12*** | (0.08, 0.15) | 3.92*** | (2.68, 5.16) | 0.41** | (0.14, 0.68) | -1.47** | (-2.34, -0.59) |
| | 45 to 54 | 0.06 | (-0.01, 0.12) | 4.56*** | (3.52, 5.60) | 0.38** | (0.15, 0.60) | -0.24 | (-1.12, 0.63) |
| | 55 to 64 | 0.07 | (0.00, 0.14) | 3.53*** | (2.34, 4.72) | 0.36** | (0.12, 0.59) | -0.21 | (-0.90, 0.48) |
| | 65+ | 0.04 | (-0.02, 0.10) | 1.58*** | (0.68, 2.48) | 0.20*** | (0.10, 0.31) | -0.45 | (-1.05, 0.14) |
| Education Group | | | | | | | | | |
| | College grad | 0.11*** | (0.06, 0.16) | 3.40*** | (2.49, 4.31) | 0.23*** | (0.13, 0.33) | -1.30*** | (-1.70, -0.89) |
| | Not a college grad | 0.04** | (0.02, 0.07) | 3.05*** | (2.22, 3.88) | 0.40*** | (0.28, 0.52) | -0.62*** | (-0.93, -0.31) |
| Employment Group | | | | | | | | | |
| | Unemployed | 0.18** | (0.07, 0.29) | 5.73*** | (4.03, 7.42) | 0.40 | (-0.04, 0.84) | -3.55*** | (-5.17, -1.93) |
| | Employed | 0.10*** | (0.05, 0.15) | 3.89*** | (2.84, 4.94) | 0.32*** | (0.19, 0.46) | -1.37*** | (-1.80, -0.94) |
| | Not in labor force | 0.06** | (0.02, 0.10) | 1.89*** | (0.96, 2.82) | 0.18** | (0.06, 0.31) | -0.62* | (-1.15, -0.08) |
| Student Status | | | | | | | | | |
| | Student | 0.13** | (0.03, 0.23) | 1.21 | (-0.22, 2.65) | -0.06 | (-0.3, 0.17) | -0.99** | (-1.74, -0.25) |
| | Not a student | 0.09*** | (0.05, 0.13) | 3.42*** | (2.61, 4.23) | 0.29*** | (0.2, 0.38) | -1.15*** | (-1.42, -0.88) |

**Source:** Authors' analysis of data from the 2016–2020 Behavioral Risk Factor Surveillance System (BRFSS). **Notes:** The exhibit displays regression-adjusted estimates for the number of sleep hours per day, and whether the respondent reported participating in any physical activities or exercise during the past month. All models were accounted for BRFSS post-stratification weights, with standard errors clustered by state

*p< .05

**p < .01

***p< .001.

respondents (+4.06 pp, 95% CI: 3.35 to 4.77). In contrast, mean sleep hours did not change significantly for respondents in households earning under $15,000, those who identify as Hispanic, multiracial, or other race, and students.

The mean number of days during the past thirty when alcohol was consumed increased at a national level (+0.27 days, 95% CI: 0.18 to 0.37). Most subgroups also experienced increases,

especially respondents who are female (+0.40 days, 95% CI: 0.31 to 0.48), identify as White (+0.38 days, 95% CI: 0.27 to 0.49) or multiracial (+0.49 days, 95% CI: 0.17 to 0.80), those aged 35–44 (+0.41 days, 95% CI: 0.14 to 0.68), and those who did not graduate college (+0.40 days, 95% CI: 0.28 to 0.52). In contrast, alcohol consumption days did not change for households earning between $15,000 and $35,000 per year; respondents who identify as Black, Hispanic, or other race; those aged 18–24; and those who were unemployed or students.

Lastly, the percentage of Americans who identify as current smokers decreased during the pandemic (-1.11 pp, 95% CI: -1.39 to -0.83). Most population subgroups experienced declines in smoking, especially among respondents who were unemployed (-3.55pp, 95% CI: -5.17 to -1.93), aged 25–34 (-2.34pp, 95% CI: -2.92 to -1.77) or 18–24 (-1.98pp, 95% CI: -2.81 to -1.14), identify as Hispanic (-2.14pp, 95% CI: -3.09 to -1.18) or Black (-1.58pp, 95% CI: -2.40 to -0.75), and households earning $15,000 to $25,000 (-1.93pp, 95% CI: -2.55 to -1.31). No significant changes were observed for respondents who were aged 45+, identify as other race or multiracial, or households earning either less than $15,000 or $35,000 to $5,000.

## Discussion

The COVID-19 pandemic dramatically changed the day-to-day lives of Americans, and our results expand our knowledge about the impact of the COVID-19 pandemic on self-reported health status and engagement in healthy behaviors. At a national level, we observed a mean increase of 0.14 days per person per month spent in poor mental health during March-December 2020. These findings are in line with studies on the negative psychological consequences of previous infectious disease outbreaks [18–22]. We also observed substantial heterogeneity by population subgroup. For instance, women experienced +0.29 days in poor mental health, while no significant effect was observed for men. Several of our results disagree with previously published findings; respondents who were White, employed, higher income, middle-aged, and did not have a college degree experienced the largest increases in poor mental health days. A discussion of the shared and unique challenges faced by each population group during the pandemic is outside the scope of this work. However, prior studies have found that employed individuals experienced heightened fears of COVID-19 exposure and contagion of themselves and their families, inability to secure childcare due to school closures, job insecurity, or difficulty transitioning to remote work during shelter-in-place orders [23–26]. Furthermore, prior studies often assessed mental health of population subgroups at a single point in time, without controlling for pre-existing trends. For instance, our results comport with prior research using Healthy Minds study data from 2013 to 2021; the authors concluded that the increased prevalence of mental health problems among college students during COVID-19 represented a continuation of pre-existing time trends rather than a unique spike [9,10]. Indeed, our results in Fig 1 suggests the decline in poor mental health days was a temporary deviation from an increasing national trend.

Despite the widespread disruption of in-person medical services during the COVID-19 pandemic, the population-level number of days spent in poor physical health also decreased during the pandemic. In fact, every demographic group experienced a decrease in poor physical health days. Correspondingly, we found that the number of days when poor health prevented usual activities decreased both overall and among most population subgroups. These results may seem counterintuitive; however, our findings comport with prior work documenting positive changes in health and health behaviors during economic downturns. Unemployment spells and economic downturns are associated with temporary reductions in smoking, alcohol consumption, obesity, physical inactivity, and air pollution [12–14]. We found that sleep and exercise behaviors improved for nearly every subgroup; increases in time spent at

home due to COVID-19 mitigation efforts (e.g., shelter in place orders) may have allowed individuals to devote more time to self-care activities [27]. Smoking status also decreased during the pandemic for most groups, although alcohol consumption increased. However, these changes may lack durability in the face of extended lockdowns and associated social isolation. Future research should examine whether our findings replicate in international contexts or represent idiosyncrasies of U.S. workplace culture.

## Study limitations

The results of this study should be interpreted in the context of the following limitations. First, the BRFSS is a cross-sectional survey. We cannot infer causality and all findings should be interpreted as associations. Second, our outcomes are self-reported; however, prior work has validated the BRFSS responses for physical activity self-reported health against data sources [28]. Third, our outcomes do not capture the severity of health challenges (e.g., minor versus major depression) or the quality of health behaviors (e.g., exercise duration). Fourth, while the BRFSS has a high response rate for phone surveys (47.9% in 2020), there is also a potential for nonresponse bias. However, we utilized BRFSS survey weights based on the age, gender, and racial distribution of the targeted population and multiple imputation using additive regression, bootstrapping, and predictive mean matching to adjust for the potential non-response bias [16,17,29]. Lastly, the BRFSS core survey does not contain measures of social engagement, nutrition or access to behavioral health services that may have important implications for physical and mental health. Despite these limitations, our findings have important policy implications and emphasize the need for sustained social and financial assistance to mitigate the pandemic's adverse impacts on mental health.

## Conclusion

In conclusion, we find the onset of COVID-19 was associated with positive effects on self-reported physical health, hours of sleep per day, exercise participation, and smoking status. In contrast, alcohol consumption increased, and the pandemic had heterogeneous effects on the number of days in poor mental health or when poor health prevented usual activities. Taken together, our results also emphasize the importance of targeted outreach and interventions to improve mental health in those who may be disproportionately affected by the pandemic and to experience significant distress.

## Supporting information

**S1 File.**
(DOCX)

## Author Contributions

**Conceptualization:** Kevin N. Griffith.

**Data curation:** Madison Hooper, Morgan Reinhart, Kevin N. Griffith.

**Formal analysis:** Morgan Reinhart, Kevin N. Griffith.

**Funding acquisition:** Kevin N. Griffith.

**Investigation:** Kevin N. Griffith.

**Methodology:** Madison Hooper, Kevin N. Griffith.

**Project administration:** Kevin N. Griffith.

**Supervision:** Stacie B. Dusetzina, Colin Walsh, Kevin N. Griffith.

**Validation:** Kevin N. Griffith.

**Writing – original draft:** Madison Hooper, Morgan Reinhart, Stacie B. Dusetzina, Colin Walsh, Kevin N. Griffith.

**Writing – review & editing:** Madison Hooper, Stacie B. Dusetzina, Colin Walsh, Kevin N. Griffith.

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
