## [Decision Letter · Decision Letter 0]

6 Jun 2023

PONE-D-22-34054Trends in U.S. self-reported health and self-care behaviors during the COVID-19 pandemicPLOS ONE

Dear Dr. Griffith,

Thank you for submitting your manuscript to PLOS ONE. After careful consideration, we feel that it has merit but does not fully meet PLOS ONE’s publication criteria as it currently stands. Therefore, we invite you to submit a revised version of the manuscript that addresses the points raised during the review process.

We look forward to receiving your revised manuscript.

Kind regards,

Yohannes Kebede, Ph.D.

Guest Editor

PLOS ONE

Journal Requirements:

“KNG's effort was supported in part by a grant from the U.S. Agency for Healthcare Research & Quality (https://www.ahrq.gov/, K12 HS026395).”

4. Please amend the manuscript submission data (via Edit Submission) to include author Morgan Reinhart.

Additional Editor Comments (if provided):

My additional comments

Clear indicate in the limitation subsection the following points:

1. Measurements of outcomes are only based on days without considering "extents: e.g. perceieved severity and frequency were one of the missed good measurement indicator for health outcomes like sleep, exercise, etc

2. Self reported health outcomes overly represented by sleep and exercise while omitting access to counseling and communication services, nutrition, engagement in discussions, etc during Covid-19

3. Measurements of self reports should have better done by scales

Add the following in the discussion section:

1. Why high income is associated to poor health outcome? Check them against two things: 1) whether or not measurement gaps exist, 2) if there has been extra expectation experienced by those who have higher income that could link to poor health outcomes

Reviewers' comments:

Reviewer's Responses to Questions

**Comments to the Author**

1. Is the manuscript technically sound, and do the data support the conclusions?

Reviewer #1: Yes

Reviewer #2: Partly

2. Has the statistical analysis been performed appropriately and rigorously? 

Reviewer #1: Yes

Reviewer #2: Yes

3. Have the authors made all data underlying the findings in their manuscript fully available?

Reviewer #1: Yes

Reviewer #2: Yes

4. Is the manuscript presented in an intelligible fashion and written in standard English?

Reviewer #1: Yes

Reviewer #2: Yes

5. Review Comments to the Author

Reviewer #1: Review: PONE-D-22-34054

Reviewer comments to the authors

Summary

The authors conduct a study of changes in health and health behaviors during the first year of the COVID-19 pandemic in the United States using the Behavior Risk Factor Surveillance System (BRFSS) as the data source. The authors examine mental health, days in poor physical health, and activity days to determine how the pandemic, associated shutdowns, and stress might have changed responses to BRFSS questions using two-tailed t-tests and interrupted time series regression models. The authors find that respondents reported increases in the days in poor mental health but also increases in sleep and exercise. Results were heterogenous and varied based several socioeconomic indicators including race, income, and education. The paper is very well-written and is a suitable fit for PLOS One. I have a few questions and comments for the authors.

1.In the discussion, I think the authors could discuss their findings a bit more and perhaps the implications for mental health and physical health of the US population. Findings might also be linked to public health initiatives or policy. In addition, were any findings surprising or unexpected (such as increase in physical health among all respondents, which seems to be the only consistent demography response).

2.Also in the discussion, the authors might discuss how the increases in sleep and physical activity might be related to the increase in time available for such activities for those populations. What does that say about workplace culture and life in the US? While the authors are correct in restricting their discussion to associations only, I think some speculation or perhaps a further research section might allow them to engage with the “why did we find this?” portion of this paper that is somewhat missing.

3.Figure 1 is very good and shows a distinct trend in poor mental health days increasing dramatically in 2020, but the increase is linked to an initial decrease in early 2020. Further, looking backward, from 2016 to 2019, there was a steady rise in poor mental health days. If the pandemic had not happened, would poor mental health days still have increased (it seems like, visually at least, that the increase was happening prior to the pandemic). It might be worth considering a comparison of the slopes for the change over time before and after the pandemic. Perhaps COVID-19 merely brought attention to an issue that has been building in the US behind the scenes.

4.To me, Figure one shows the strongest association for changes in physical health, which is a very interesting finding since many people speculate that physical health deteriorated for many during the pandemic. Your study would suggest otherwise and the discussion should state this and suggest why.

Reviewer #2: This is a valuable article and can add input to the existing literature. However, there are some points that need clarity. Please find the comments below:

Comments

Title:

1. The term ‘self-reported health’ is not informative. Do the researchers want to say ‘self-reported health impacts’? Please revise it. In addition, it is also attractive to say ‘engagement in healthy behaviors’ instead of ‘self-care behaviors’.

2. It is not good to use abbreviations or acronyms in the title. Hence, instead of the Acronym ‘U.S.’ write in full words ‘United States’.

Abstract:

Importance: However, it remains unclear how the COVID-19 pandemic impacted self-reported mental and physical health and influence human behaviors.

Objective: please revise it. To assess changes in self-reported health outcomes/impacts during the COVID-19 pandemic and trends of engagement in health behaviors.

In abstract section; good to include the summary of analyses that researchers carried out.

Introduction:

1. The first sentence needs revision. The novel coronavirus (COVID-19) began to spread across the world in December 2019 and on March 13, 2020. In response to the pandemic, different countries of the world had been taking different prevention and control measures. Similarly, the United States declared a national public health emergency.

2. The last paragraph of the introduction needs revision. Indeed, the researchers should explain the gaps in detail and convince why this study is needed and its input to the existing literature.

3. Our objectives were........no need of making separate paragraphs for the objective. In addition, no need of listing them. Rather write it in a single sentence and merge it with the last paragraph of the introduction section.

Methods:

1. Some variables need Operational definitions and explain how they were measured. Please define poor mental health, poor physical health, and exercise. Explain in detail how these variables were measured.

2. Did the researchers check assumptions for the statistical tests carried out? What are the findings of assumptions? Any violation?

Results

1. The researchers showed the trends in poor physical and mental health days (figure 1). That is good. Would you please explain the trends of self-care behaviors/engagement in health behaviors during the COVID-19 pandemic?

Discussion

The discussion is shallow. The researchers should discuss the pertinent findings from both theoretical and empirical perspectives.

6. PLOS authors have the option to publish the peer review history of their article (what does this mean?). If published, this will include your full peer review and any attached files.

Reviewer #1: **Yes: **Dustin T. Hill

Reviewer #2: No

---

## [Author Response · Author response to Decision Letter 0]

6 Aug 2023

Please see attached response letter.

---

## [Decision Letter · Decision Letter 1]

22 Aug 2023

PONE-D-22-34054R1Trends in U.S. self-reported health and self-care behaviors during the COVID-19 pandemicPLOS ONE

Dear Dr. Griffith,

Thank you for submitting your manuscript to PLOS ONE. After careful consideration, we feel that it has merit but does not fully meet PLOS ONE’s publication criteria as it currently stands. Therefore, we invite you to submit a revised version of the manuscript that addresses the points raised during the review process.

We look forward to receiving your revised manuscript.

Kind regards,

Yohannes Kebede, Ph.D.

Guest Editor

PLOS ONE

Journal Requirements:

Additional Editor Comments:

Please address the minor comments provided by reviewers.

Reviewers' comments:

Reviewer's Responses to Questions

**Comments to the Author**

1. If the authors have adequately addressed your comments raised in a previous round of review and you feel that this manuscript is now acceptable for publication, you may indicate that here to bypass the “Comments to the Author” section, enter your conflict of interest statement in the “Confidential to Editor” section, and submit your "Accept" recommendation.

Reviewer #1: All comments have been addressed

Reviewer #2: (No Response)

2. Is the manuscript technically sound, and do the data support the conclusions?

Reviewer #1: Yes

Reviewer #2: Yes

3. Has the statistical analysis been performed appropriately and rigorously? 

Reviewer #1: Yes

Reviewer #2: Yes

4. Have the authors made all data underlying the findings in their manuscript fully available?

Reviewer #1: Yes

Reviewer #2: Yes

5. Is the manuscript presented in an intelligible fashion and written in standard English?

Reviewer #1: Yes

Reviewer #2: Yes

6. Review Comments to the Author

Reviewer #1: The authors have made the requested changes. The resulting manuscript has much better flow and easier to follow introduction. I have no additional comments to add. I appreciate the authors’ time and effort on the revisions, and I think the paper is acceptable to publish.

Reviewer #2: The authors addressed most of the comments and well improved the document. However, the following two main points still need clarification/explanation to strengthen the rigorousness of the article.

Methods section:

1. Some variables such as poor mental health, poor physical health and exercise need to be defined (operational definition) and explain in detail how they were measured.

2. Please describe the assumptions that the researchers checked for the statistical tests carried out? Please incorporate the findings of assumptions in the method section.

7. PLOS authors have the option to publish the peer review history of their article (what does this mean?). If published, this will include your full peer review and any attached files.

Reviewer #1: **Yes: **Dustin T. Hill

Reviewer #2: No

---

## [Author Response · Author response to Decision Letter 1]

31 Aug 2023

Dear Dr. Kebede,

Thank you for the opportunity to revise and resubmit our manuscript, “Trends in U.S. self-reported health and self-care behaviors during the COVID-19 pandemic” for consideration at PLOS ONE. The reviewers’ and editor’s comments have been very helpful and strengthened the paper. Our responses are below. 

Sincerely,

Kevin N. Griffith, Ph.D.

Department of Health Policy

Vanderbilt University School of Medicine

Reviewer #2 comments:

1) Some variables such as poor mental health, poor physical health and exercise need to be defined (operational definition) and explain in detail how they were measured.

Thanks for the opportunity to clarify. All outcomes were defined in Appendix A1 including the exact survey text and response format; we have moved this to the main text (Table 1) to aid the reader.

2) Please describe the assumptions that the researchers checked for the statistical tests carried out. Please incorporate the findings of assumptions in the method section.

Thank you for this question. We estimated ordinary least squares regressions (OLS) for all analyses. We estimated a total of 182 regression models (26 population groups x 7 outcomes) and it would not be possible to present all of the diagnostic regression output. There are 5 key assumptions for OLS:

i) Linearity of regressors: All of our independent variables are binary, thus this assumption does not need to be checked. 

ii) Normality of model residuals. As stated in the Methods section: “Due to the large sample size and our interest in population average effects, we opted to conduct linear probability models using ordinary least squares to aid in interpretability of results.” Further, we include the following citation which describes how the normality assumption is not important for analyses of large public health datasets: Lumley T, Diehr P, Emerson S, Chen L. The importance of the normality assumption in large public health data sets. Annu Rev Public Health. 2002;23. doi:10.1146/annurev.publhealth.23.100901.140546

iii) Homoscedasticity of model residuals. We also state in the Methods section: “We employed heteroskedasticity-robust standard errors in all regression models.” This obviates the need to check this assumption.

iv) Independence of observations. The BRFSS is a repeated cross-sectional survey with a new sample of respondents each year. The primary risk here is that residents in the same state are subjected to the same set of laws and policies (e.g., shelter in place orders). Thus, our Methods section includes the following which addresses this concern: “Errors were also clustered by state to account for pandemic-related policies that may affect their residents.”

v) No multicollinearity. Multicollinearity is usually obvious to diagnose because the standard errors of your regression explode if it is present. However, we also checked the variance inflation factor (VIF) for each regression specification. The VIF did not exceed 10 in any of our models, thus multicollinearity was not an issue.

We defer to the editor for whether this information should be included in the manuscript text versus the peer review tab, since the former is unusual.

---

## [Editor Report · Decision Letter 2]

4 Sep 2023

Trends in U.S. self-reported health and self-care behaviors during the COVID-19 pandemic

PONE-D-22-34054R2

Dear Dr. Griffith,

We’re pleased to inform you that your manuscript has been judged scientifically suitable for publication and will be formally accepted for publication once it meets all outstanding technical requirements.

Kind regards,

Yohannes Kebede, Ph.D.

Guest Editor

PLOS ONE
---

## [Editor Report · Acceptance letter]

11 Sep 2023

PONE-D-22-34054R2 

Trends in U.S. self-reported health and self-care behaviors during the COVID-19 pandemic 

Dear Dr. Griffith:

I'm pleased to inform you that your manuscript has been deemed suitable for publication in PLOS ONE. Congratulations! Your manuscript is now with our production department. 

Kind regards, 

on behalf of

Dr. Yohannes Kebede 

Guest Editor

PLOS ONE